# Effects of Individual and Simultaneous Selenium and Iodine Biofortification of Baby-Leaf Lettuce Plants Grown in Two Different Hydroponic Systems

Martina Puccinelli [1,*], Fernando Malorgio [1,2], Luca Incrocci [1], Irene Rosellini [3] and Beatrice Pezzarossa [3]

1 Department of Agriculture, Food and Environment, University of Pisa, Via del Borghetto 80, 56124 Pisa, Italy; fernando.malorgio@unipi.it (F.M.); luca.incrocci@unipi.it (L.I.)

2 Interdepartmental Research Center Nutraceuticals and Food for Health, University of Pisa, Via del Borghetto 80, 56124 Pisa, Italy

3 Research Institute on Terrestrial Ecosystems, National Research Council, Via G. Moruzzi 1, 56124 Pisa, Italy; irene.rosellini@cnr.it (I.R.); beatrice.pezzarossa@cnr.it (B.P.)

* Correspondence: martina.puccinelli@agr.unipi.it

**Abstract:** The iodine (I) and selenium (Se) deficiencies affect approximately 30% and 15%, respectively, of the global population. The biofortification of vegetables is a valid way to increase the intake of iodine and selenium through the diet. This study was carried out on baby-leaf lettuce to investigate the effects on plant growth, leaf quality, and leaf I and Se accumulation of adding potassium iodide and sodium selenate, separately and simultaneously, to the nutrient solution in a floating system and aeroponics. The effect of I and Se biofortification on post-harvest quality of lettuce leaves was also evaluated. Our results evidenced that the Se and I treatments increased the content of the two microelements in lettuce leaves without any negative interactions in the plants, when applied either separately or simultaneously. Both hydroponic systems proved to be suitable for producing Se and/or I enriched lettuce. Biofortification with Se was more effective when performed in aeroponics, whereas I biofortification was more effective in the floating system. Quality of leaves during post-harvest storage was not affected by neither of the treatments. Lettuce leaves enriched with 13 μM Se and 5 μM I could be good dietary sources of Se and I without inducing toxic effects in humans.

**Keywords:** aeroponics; floating system; *Lactuca sativa*; potassium iodide; sodium selenate

## 1. Introduction

Iodine (I) and selenium (Se) deficiency affects approximately 30% and 15%, respectively, of the global population [1].

Iodine is an essential element for human health as it is involved in the synthesis of the thyroid hormone [2]. The adequate intake (AI) is 150 μg I day$^{-1}$ [3], and I deficiency is one of the most widespread micronutrient deprivations, causing an insufficient secretion of thyroid hormones that leads to a series of iodine deficiency disorders (IDDs) [4].

Selenium is an essential trace element in animals and humans. As a component of seleno-amino acids and selenoproteins, and as a cofactor of glutathione peroxidase (GSH-Px; EC 1.11.1.9) [5], Se is involved in various metabolic processes such as thyroid hormone metabolism, antioxidant defense, and immune function [5]. Selenium is an antioxidant and may have a positive effect on plant metabolism [6] and delay plant senescence by reducing ethylene production [7]. In leaves of lettuce plants, Se may have a positive effect in reducing the nitrate content [8] and in increasing the phenolic acid content [9] and the pigments [10]. The AI of selenium proposed by EFSA [11] for adult men and women is 70 μg day$^{-1}$.

Deficient iodine and selenium status can be alleviated by dietary supplements. Salt iodization has contributed to reducing iodine deficiency in many countries. However, the

loss of I occurring during storage, transportation, and processing [2], as well as the implementation of policies aimed at reducing salt intake in order to prevent hypertension and cardiovascular diseases, mean that salt ionization is still inadequate, and I biofortification of vegetables might be a viable option to fight IDDs [12].

The biofortification of vegetables increases the human intake of iodine [12] and selenium [13] by improving the nutritional quality of fruit and vegetables during plant growth, rather than during post-harvest handling and storage [14].

The simultaneous application of Se and I would provide two essential elements at the same time, promoting the more efficient intake of Se and I through the diet. Nevertheless, since I and Se are not essential mineral nutrients for plants, the simultaneous use for biofortification of even small doses may induce toxicity in plants [15].

Several works have been conducted in hydroponics on iodine [16–18] and selenium enrichment [7,8,19]; however, the biofortification of lettuce through the interaction between the two elements has not been extensively studied. The simultaneous application of Se and I has been investigated only in spinach [20] and lettuce [21–23].

Closed-loop hydroponic cultivation facilitates the production of biofortified vegetables, obtaining a higher crop yield, higher quality produce, lower consumption of water and fertilizers, and the reduced release of pesticides into the environment [24]. The floating system and aeroponics are two hydroponics techniques that are suitable for cultivating leafy vegetables, including lettuce [18]. Floating systems, in which plant roots are submerged in the nutrient solution, are widely used to produce baby leaf vegetables, with a short cycle and high plant density. In aeroponics, plants roots are suspended in the air and frequently sprayed with the nutrient solution [24]. Compared to floating system, where the roots can only use the oxygen dissolved in the nutrient solution, in aeroponics roots have greater access to oxygen, reducing the risk of root hypoxia [25].

Lettuce (*Lactuca sativa* L.) is widely cultivated to produce baby leaves as well as mature heads of leaves. The detached leaves need to be rapidly protected against deterioration during storage. Ready-to-eat immature leaves of leafy vegetables are healthy and convenient to eat, and thus are particularly valued by consumers [26].

Since there is a lack of knowledge on the use of aeroponics for the biofortification of vegetables with I and Se, this study was carried out on baby-leaf lettuce to evaluate the effects of two hydroponic techniques (floating system and aeroponics) and the addition of potassium iodide (KI) and sodium selenate ($Na_2SeO_4$) to the nutrient solution, both separately and simultaneously, on plant growth, leaf quality, and leaf I and Se accumulation. The possible effect of I and Se biofortification on the post-harvest quality of lettuce leaves was also evaluated.

## 2. Materials and Methods

### 2.1. Plant Materials and Growing Conditions

Two experiments were carried out under greenhouse conditions at the University of Pisa, Italy (lat. 42′42″48 N, long. 10°24′52″92 E), in autumn 2019 and 2020.

Lettuce (*Lactuca sativa* var. crispa L. "Salad Bowl") seeds were sown in 240-cell plug-trays filled with rockwool and vermiculite, and the trays were placed in a growth chamber at 25 °C for 5 days. Ten days after sowing, lettuce seedlings were transplanted to aeroponics and floating systems. The aeroponics was made up of four separate system closed at the top with eight polystyrene panel, with a total volume of 220 L m$^{-2}$. In total, 160 plants were placed in each system. The mixing tank of each system contained 200 L of nutrient solution, which was sprayed on the plant roots for 20 s every 5 min during the day, and every 40 min during the night.

The floating system was made up of eight separate 50 L plastic tanks (water depth 25 cm) with a polystyrene tray hosting 16 plants of lettuce.

In both hydroponic systems, the crop density was 100 plants m$^{-2}$ on a ground area basis.

All the plants were supplied with a nutrient solution containing N-NO$_3^-$ 14.0 mM, N-NH$_4^+$ 2.0 mM, P 2.0 mM, K 10.0 mM, Ca 4.5 mM, Mg 2.0 mM, S-SO$_4$ 5.0 mM, Fe 40.0 µM,

B 40.0 μM, Cu 3.0 μM, Zn 10.0 μM, Mn 10.0 μM, and Mo 1.0 μM. The pH and electrical conductivity (EC) values were 5.6 and 2.32 dS m$^{-1}$, respectively. The pH and EC were checked every day and remained within 10% of the values measured in the newly prepared nutrient solution.

In the floating system, the nutrient solution was continuously aerated, and the oxygen content ranged between 4 and 6 g m$^{-3}$ with an average of 5 mg L$^{-1}$. In the aeroponics system, the oxygen content of the nutrient solution in the mixing tank was always higher than 8.5 mg L$^{-1}$ throughout the entire experiment.

During the experiments, the nutrient solution was never replaced or reintegrated.

Climatic conditions were continuously monitored by a weather station located inside the greenhouse. The mean air temperature was 22 and 20 °C, and daily solar radiation was 5.44 and 4.83 MJ m$^{-2}$ day$^{-1}$, during the first and second experiment, respectively. The climatic and cultivation parameters are shown in Tables S1 (first experiment) and S2 (second experiment) of the Supplementary Materials.

### 2.2. Experimental Design

The treatments consisted of the combination of three factors: the hydroponic system, and the concentrations of Se and I in the nutrient solution. Three replicates were realized by randomly choosing 8 plants for each treatment in each hydroponic unit.

The Se and I treatments started seven days after plants have been transplanted in the hydroponic system. Three replications, constituted by 8 plants, were used for each treatment (control, Se, I, and Se + I) in both cultivation systems. Se was added to the nutrient solution as sodium selenate (Na$_2$SeO$_4$), at a concentration of 13 μM (1 mg Se L$^{-1}$), and/or I, as potassium iodide (KI), at a concentration of 5 μM (0.2 mg I L$^{-1}$). These concentrations were chosen because previous results, obtained in experiments with I [18] or Se [7], suggested that they could significantly increase the content of I and Se in leafy vegetables without inducing toxicity. Control plants received neither selenium nor iodine.

### 2.3. Post-Harvest Storage

During the second experiment, 10 g FW of fully expanded lettuce leaves were packaged in 750 mL plastic trays and stored at 4 °C and 24 h of white LED light for 7 days. Three plastic trays were used for each treatment and each sampling time, i.e., after 2 (T2), 4 (T4), and 7 (T7) days of storage, for a total of 72 plastic trays.

### 2.4. Determinations

#### 2.4.1. Biomass Production

Lettuce was harvested 17 days after transplanting by cutting the plants 2 cm above the collar, and fresh weight (FW) of 8 plants for each replicate was determined. After drying in a ventilated oven at 50 °C to constant weight, the dry weight (DW) was measured. The fresh and dry biomass production were expressed as g FW m$^{-2}$ and g DW m$^{-2}$, respectively.

During post-harvest storage, to calculate the weight reduction, we measured the FW of lettuce leaves and compared them with the initial weight (10 g).

#### 2.4.2. Selenium, Iodine, and Nitrate Analyses

Total selenium content was determined in oven-dried ground leaf samples after digestion with nitric and perchloric acids and reduction by hydrochloric acid [27]. The digests were analyzed by an atomic absorption spectrometer (SpectrAA 240FS, Varian Inc., Mulgrave, VIC, Australia) coupled with a hydride generation system (VGA 77, Varian Inc., Mulgrave, VIC, Australia).

Total iodine content was determined in oven-dried ground leaf samples using the UNI EN 15111:2007 method [28] for the mineralization and analysis. Three replicates were analyzed for each treatment. Analyses were carried out by the CAIM group (Follonica, GR, Italy).

The nitrate content in the lettuce leaves was measured spectrophotometrically in dry samples, after extraction with distilled water (100 mg DW in 20 mL) at room temperature for 2 h, using the salicylic–sulfuric acid method [29].

2.4.3. Leaf Photosynthetic Pigments, Flavonoids, Total Phenols, and Total Antioxidant Capacity

The content of chlorophyll, carotenoid, flavonoid, total phenols, and the antioxidant capacity were analyzed in the fully expanded lettuce leaves of plants from three trays for each treatment at harvest time (T0) and at each sampling time (T2, T4 and T7) during post-harvest.

To measure the chlorophylls and carotenoid content, we cut foliar fresh tissues in small discs, and each sample (100 mg) was added with 5 mL methanol 99% $v/v$, then extracted by sonication and maintained for 24 h at 4 °C. The methanol extract was used to spectrophotometrically determine the concentrations of chlorophyll a, chlorophyll b, and carotenoids using the equation reported by Welburn and Lichtenthaler [30].

To determine the flavonoid content, we added 0.1 mL of the methanol extract to 0.06 mL of $NaNO_2$ (5%); 0.04 mL of $AlCl_3$ (10%); and after five minutes, 0.4 mL of NaOH and 0.2 mL of $H_2O$. The absorbance was then read at 510 nm. The results were expressed as mg catechin $g^{-1}$ FW [31].

The same methanol extracts were used to determine the total phenol content using the Folin–Ciocalteau reagent according to Kang and Saltveit [32]. The total phenol content was calculated using the calibration curve containing 0, 50, 100, 150, and 250 mg gallic acid $L^{-1}$; values were expressed as mg of gallic acid (GAE) $g^{-1}$ FW.

The total antioxidant capacity was measured with the ferric reducing ability of plasma (FRAP) assay [33], using an aliquot of methanol extract, used for the determination of pigments and phenol content. The results were expressed as μmol Fe(II) $mg^{-1}$ FW.

*2.5. Contribution to Se and I Dietary Intake and Health Risk Assessment*

The estimated dietary intakes (EDI, μg $day^{-1}$) of Se and I, following the consumption of 100 g of lettuce leaves, were calculated according to the following equation:

$$EDI = C \times SP/1000$$

where

C = Se or I concentration (μg $kg^{-1}$ FW) in the lettuce leaves
SP = a serving size of 100 g of lettuce leaves.

In the evaluation of the contribution of the Se- and I-enriched lettuce leaves to human Se and I needs, EDI was expressed as the percentage (EDI%) of the adequate intake (AI) of Se (70 μg $day^{-1}$) [11] or I (150 μg $day^{-1}$), for an adult [3].

To assess the possible health risk due to the intake of Se and I provided by the consumption of biofortified lettuce leaves, we calculated the health risk index (HRI) as follows:

$$HRI = EDI/UL$$

where:

EDI = as defined above;
UL = tolerable upper intake level of Se (300 μg $day^{-1}$) and I (600 μg $day^{-1}$) [34].

In general, UL is the maximum chronic daily intake of a nutrient (from all sources) that is assumed to not induce an appreciable risk of adverse health effects to humans [34].

*2.6. Statistical Analysis*

Data were tested for homogeneity of error variances with Levene's test, and subsequently were subjected to three-way ANOVA, with I and Se treatment and hydroponic system as variables. Data from the second experiment, regarding post-harvest storage,

were also subjected to three-way ANOVA with I and Se treatments and time of storage as variables. Mean values were separated by Duncan's post hoc test ($p < 0.05$). Statistical analysis was performed using R Statistical Software.

## 3. Results

Unless otherwise stated, the results reported in the following sections are the average values.

### 3.1. Selenium and Iodine Content

Separate addition of iodine and selenium to the nutrient solution resulted in a significant increase in the leaf content of I or Se, respectively. The I concentration in I-treated plants ranged from 65.4 to 86.3 mg kg$^{-1}$ DW in the first experiment (Table 1) and from 91.9 to 140.5 mg kg$^{-1}$ DW in the second experiment (Table 2). The Se concentration in Se-treated plants ranged from 3.3 to 6.5 mg kg$^{-1}$ DW in the first experiment (Table 1) and from 4.2 to 5.2 mg kg$^{-1}$ DW in the second experiment (Table 2).

**Table 1.** Iodine (I) and selenium (Se) content, fresh (FW) and dry (DW) weight, dry matter percentage (DW/FW), and nitrate (NO$_3$) content in leaves of lettuce plants grown in two different hydroponic systems (HS), floating (FS) and aeroponics (AE), and at different concentrations of I and Se in the nutrient solution, in the first experiment.

| Hydroponic System | I Added (μM) | Se Added (μM) | I (mg kg$^{-1}$ DW) | Se (mg kg$^{-1}$ DW) | FW (g m$^{-2}$) | DW (g m$^{-2}$) | DW/FW (%) | NO$_3$ (mg kg$^{-1}$ FW) |
|---|---|---|---|---|---|---|---|---|
| FS | 0 | 0 | 4.70 | 0.09 | 174.2 c | 6.3 | 3.63 | 924.7 |
| | | 13 | 2.46 | 3.33 | 198.3 c | 7.9 | 3.99 | 1135.8 |
| | 5 | 0 | 86.30 | 0.03 | 165.8 c | 6.6 | 3.99 | 1066.0 |
| | | 13 | 72.55 | 3.44 | 193.8 c | 5.7 | 2.92 | 755.6 |
| A | 0 | 0 | 4.02 | 0.10 | 478.1 b | 16.5 | 3.49 | 1012.5 |
| | | 13 | 2.45 | 6.54 | 735.0 a | 20.9 | 2.84 | 499.4 |
| | 5 | 0 | 65.43 | 0.15 | 598.3 b | 19.6 | 3.31 | 1248.1 |
| | | 13 | 71.90 | 6.46 | 538.4 b | 17.2 | 3.24 | 601.1 |
| Mean effect | | | | | | | | |
| FS | | | 41.50 | 1.72 b | 183.0 b | 6.6 b | 3.63 | 970.5 a |
| AE | | | 35.95 | 3.31 a | 587.5 a | 18.6 a | 3.22 | 840.3 b |
| | 0 | | 3.37 b | 2.67 | 438.4 | 14.1 | 3.42 | 865.7 |
| | 5 | | 72.97 a | 2.68 | 412.9 | 13.5 | 3.35 | 919.1 |
| | | 0 | 39.03 | 0.10 b | 390.9 b | 13.4 | 3.56 | 1076.3 a |
| | | 13 | 37.3 | 5.25 a | 460.4 a | 14.1 | 3.21 | 708.5 b |
| FS | 0 | | 3.58 | 1.71 | 186.3 | 7.1 | 3.81 | 1030.2 a |
| | 5 | | 79.43 | 1.73 | 179.8 | 6.1 | 3.46 | 910.8 ab |
| AE | 0 | | 3.23 | 3.32 | 606.5 | 18.7 | 3.17 | 756.0 b |
| | 5 | | 68.67 | 3.31 | 568.4 | 18.4 | 3.27 | 924.6 ab |
| FS | | 0 | 45.5 | 0.06 c | 170.0 c | 6.5 | 3.81 | 995.4 a |
| | | 13 | 37.51 | 3.38 b | 196.0 c | 6.8 | 3.46 | 945.8 a |
| AE | | 0 | 24.37 | 0.13 c | 538.2 b | 18.1 | 3.40 | 1130.3 a |
| | | 13 | 37.18 | 6.50 a | 636.7 a | 19.0 | 3.04 | 550.4 b |
| | 0 | 0 | 4.29 | 0.09 | 356.5 b | 12.4 b | 3.55 | 977.4 ab |
| | | 13 | 2.45 | 5.25 | 520.3 a | 15.7 a | 3.30 | 753.9 bc |
| | 5 | 0 | 73.78 | 0.10 | 425.3 ab | 14.4 ab | 3.58 | 1175.3 a |
| | | 13 | 72.16 | 5.25 | 400.6 ab | 12.6 b | 3.11 | 663.0 b |
| Significance | | | | | | | | |
| HS | | | ns | *** | *** | *** | ns | * |
| I | | | *** | ns | ns | ns | ns | ns |
| Se | | | ns | *** | * | ns | ns | *** |
| HS × I | | | ns | ns | ns | ns | ns | * |
| HS × Se | | | ns | *** | * | ns | ns | *** |
| I × Se | | | ns | ns | ** | ** | ns | * |
| HS × I × Se | | | ns | ns | * | ns | ns | ns |

Means (*n* = 3) flanked by the same letter are not statistically different for *p* = 0.05 after Duncan's test. Significance level: *** $p \leq 0.001$; ** $p \leq 0.01$; * $p \leq 0.05$; ns = not significant.

**Table 2.** Iodine (I) and selenium (Se) content, fresh (FW) and dry (DW) weight, dry matter percentage (DW/FW), and nitrate ($NO_3$) content in leaves of lettuce plants grown in two different hydroponic systems (HS), floating (FS) and aeroponics (AE), and at different concentrations of I and Se in the nutrient solution, in the second experiment.

| Hydroponic System | I Added (µM) | Se Added (µM) | I (mg kg⁻¹ DW) | Se (mg kg⁻¹ DW) | FW (g m⁻²) | DW (g m⁻²) | DW/FW (%) | NO₃ (mg kg⁻¹ FW) |
|---|---|---|---|---|---|---|---|---|
| FS | 0 | 0 | 6.18 | 0.00 | 183.3 | 7.3 | 4.0 | 915.8 |
|  |  | 13 | 4.02 | 4.84 | 179.5 | 7.1 | 3.9 | 950.9 |
|  | 5 | 0 | 140.5 | 0.00 | 187.2 | 7.4 | 4.0 | 779.2 |
|  |  | 13 | 138.0 | 4.24 | 185.9 | 6.9 | 3.7 | 836.3 |
| A | 0 | 0 | 4.10 | 0.00 | 326.7 | 13.6 | 4.2 | 846.7 |
|  |  | 13 | 3.75 | 5.07 | 407.9 | 15.4 | 3.8 | 572.9 |
|  | 5 | 0 | 91.85 | 0.00 | 332.5 | 15.0 | 4.5 | 947.4 |
|  |  | 13 | 120.00 | 5.15 | 342.3 | 13.6 | 4.0 | 711.5 |
| | | | | Mean effect | | | | |
| FS |  |  | 72.17 [a] | 2.27 [b] | 184.0 [b] | 7.2 [b] | 3.9 | 870.5 [a] |
| AE |  |  | 54.92 [b] | 2.55 [a] | 352.4 [a] | 14.4 [a] | 4.1 | 769.6 [b] |
|  | 0 |  | 4.51 [b] | 2.49 | 292.9 | 11.6 | 4.0 | 799.2 |
|  | 5 |  | 122.59 [a] | 2.39 | 277.1 | 11.5 | 4.1 | 820.8 |
|  |  | 0 | 60.66 | 0.00 [b] | 271.8 [b] | 11.5 | 4.2 [a] | 877.2 [a] |
|  |  | 13 | 66.44 | 4.88 [a] | 298.2 [a] | 11.5 | 3.9 [b] | 742.8 [b] |
| FS | 0 |  | 5.10 [c] | 2.42 | 181.4 | 7.2 | 4.0 | 933.3 [a] |
|  | 5 |  | 139.25 [c] | 2.12 | 186.5 | 7.2 | 3.9 | 807.8 [ab] |
| AE | 0 |  | 3.92 [c] | 2.53 | 367.3 | 14.5 | 4.0 | 709.8 [b] |
|  | 5 |  | 105.93 [b] | 2.58 | 337.4 | 14.3 | 4.2 | 829.4 [ab] |
| FS |  | 0 | 73.34 | 0.00 [c] | 185.3 [c] | 14.3 | 4.3 | 847.5 [a] |
|  |  | 13 | 71.01 | 4.54 [b] | 182.7 [c] | 14.5 | 3.9 | 893.6 [a] |
| AE |  | 0 | 47.97 | 0.00 [c] | 329.6 [b] | 7.4 | 4.0 | 897.0 [a] |
|  |  | 13 | 61.88 | 5.11 [a] | 375.1 [a] | 7.0 | 3.8 | 642.2 [b] |
|  | 0 | 0 | 5.14 | 0.00 | 269.3 | 11.1 | 4.1 | 874.3 |
|  |  | 13 | 3.89 | 4.98 | 316.6 | 12.1 | 3.8 | 724.1 |
|  | 5 | 0 | 116.18 | 0.00 | 274.4 | 12.0 | 4.3 | 880.1 |
|  |  | 13 | 129.00 | 4.78 | 279.8 | 11.0 | 3.9 | 761.5 |
| | | | | Significance | | | | |
| HS |  |  | *** | * | *** | *** | ns | * |
| I |  |  | *** | ns | ns | ns | ns | ns |
| Se |  |  | ns | *** | * | ns | ** | ** |
| HS × I |  |  | ** | ns | ns | ns | ns | ** |
| HS × Se |  |  | ns | * | * | ns | ns | *** |
| I × Se |  |  | ns | ns | ns | ns | ns | ns |
| HS × I × Se |  |  | ns | ns | ns | ns | ns | ns |

Means ($n = 3$) flanked by the same letter are not statistically different for $p = 0.05$ after Duncan's test. Significance level: *** $p \leq 0.001$; ** $p \leq 0.01$; * $p \leq 0.05$; ns = not significant.

The leaf Se content was higher in Se-enriched plants grown in aeroponics compared to plants grown in the floating system (Tables 1 and 2). On the other hand, in the second experiment, the leaf I content was higher in I-enriched plants grown in the floating system compared to plants grown in aeroponics (Table 2).

### 3.2. Biomass Production

The fresh biomass production was on average lower in floating system than in aeroponics. The addition of selenium to the nutrient solution led to a significant increase in fresh biomass production in plants grown in aeroponics without iodine in the first (Table 1) and in the second experiment (Table 2).

In the first experiment, the dry biomass of plants enriched only with selenium was 24.6% higher compared to plants enriched with both selenium and iodine, and 26.6% higher than the control plants (Table 1).

Plants grown in aeroponics showed significantly higher fresh and dry biomass compared to plants grown in floating system both in the first (Table 1) and second experiments (Table 2).

In the second experiment, treatment with Se induced a lower dry matter content of lettuce leaves (Table 2).

### 3.3. Qualitative Characteristics of Leaves

The leaf nitrate content was lower in the lettuce plants grown in aeroponics than in the floating system when treated with selenium, both in the first (Table 1) and second experiments (Table 2), and when iodine was not added to the nutrient solution in the second experiment (Table 2).

Plants grown without iodine showed a lower leaf nitrate content when cultivated in aeroponics than in the floating system (Tables 1 and 2).

Se-enriched plants grown in aeroponics showed a lower leaf nitrate content compared to plants grown in aeroponics without Se and to plants grown in floating system with or without Se (Tables 1 and 2).

In the first experiment, the addition of Se to the nutrient solution of I-enriched plants led to a lower leaf nitrate content (Table 1) compared to the other treatments. In addition, plants grown in aeroponics showed a higher chl a/chl b ratio, total chlorophyll, and carotenoid content than in the floating system. The highest content of total chlorophyll (0.831 mg g$^{-1}$ FW) was detected in plants grown in aeroponics and treated with Se + I (Table 3).

**Table 3.** Total phenols, antioxidant capacity (FRAP), chlorophyll a to b ratio (chl a/chl b), total chlorophyll, and carotenoid contents, measured at harvest, in leaves of lettuce plants grown in two different hydroponic systems (HS), floating (FS) and aeroponics (AE), and at different concentrations of I and Se in the nutrient solution, in the first experiment.

| Hydroponic System | I Added (μM) | Se Added (μM) | Total Phenols (mg GAE g$^{-1}$ FW) | FRAP (μmol Fe (II) g$^{-1}$ FW) | chl a/chl b | Chls Tot (mg g$^{-1}$ FW) | Car (mg g$^{-1}$ FW) |
|---|---|---|---|---|---|---|---|
| FS | 0 | 0 | 4.52 | 21.21 | 2.35 | 0.645 [b] | 0.108 [e] |
|  |  | 13 | 4.45 | 20.50 | 3.07 | 0.661 [b] | 0.136 [cd] |
|  | 5 | 0 | 4.35 | 21.36 | 2.78 | 0.654 [b] | 0.121 [de] |
|  |  | 13 | 4.84 | 19.83 | 2.80 | 0.665 [b] | 0.122 [de] |
| A | 0 | 0 | 4.28 | 21.36 | 3.53 | 0.711 [b] | 0.160 [ab] |
|  |  | 13 | 4.06 | 22.44 | 3.41 | 0.639 [b] | 0.141 [bcd] |
|  | 5 | 0 | 4.42 | 25.39 | 3.36 | 0.645 [b] | 0.148 [bc] |
|  |  | 13 | 4.26 | 20.22 | 3.40 | 0.831 [a] | 0.178 [a] |
| Mean effect |  |  |  |  |  |  |  |
| FS |  |  | 4.54 | 20.73 | 2.75 [b] | 0.656 [b] | 0.122 [b] |
| AE |  |  | 4.25 | 22.35 | 3.43 [a] | 0.707 [a] | 0.157 [a] |
|  | 0 |  | 4.29 | 21.48 | 3.17 | 0.666 | 0.139 |
|  | 5 |  | 4.43 | 21.92 | 3.14 | 0.707 | 0.146 |
|  |  | 0 | 4.38 | 22.54 | 3.09 | 0.666 | 0.138 |
|  |  | 13 | 4.35 | 20.87 | 3.22 | 0.706 | 0.147 |
| FS | 0 |  | 4.49 | 20.86 | 2.71 | 0.653 | 0.122 |
|  | 5 |  | 4.59 | 20.59 | 2.79 | 0.659 | 0.121 |
| AE | 0 |  | 4.17 | 21.90 | 3.47 | 0.675 | 0.150 |
|  | 5 |  | 4.34 | 22.81 | 3.38 | 0.738 | 0.163 |
| FS |  | 0 | 4.43 | 21.28 | 2.56 | 0.649 | 0.115 |
|  |  | 13 | 4.64 | 20.17 | 2.94 | 0.663 | 0.129 |
| AE |  | 0 | 4.35 | 23.38 | 3.45 | 0.678 | 0.154 |
|  |  | 13 | 4.16 | 21.33 | 3.41 | 0.735 | 0.159 |
|  | 0 | 0 | 4.37 | 21.30 | 3.06 | 0.685 | 0.139 |
|  |  | 13 | 4.22 | 21.67 | 3.27 | 0.648 | 0.139 |
|  | 5 | 0 | 4.39 | 23.78 | 3.13 | 0.648 | 0.137 |
|  |  | 13 | 4.49 | 20.07 | 3.16 | 0.765 | 0.155 |
| Significance |  |  |  |  |  |  |  |
| HS |  |  | ns | ns | *** | * | *** |
| I |  |  | ns | ns | ns | ns | ns |
| Se |  |  | ns | ns | ns | ns | ns |
| HS × I |  |  | ns | ns | ns | ns | ns |
| HS × Se |  |  | ns | ns | ns | ns | ns |
| I × Se |  |  | ns | ns | ns | ** | ns |
| HS × I × Se |  |  | ns | ns | ns | ** | ** |

Means (n = 3) flanked by the same letter are not statistically different for p = 0.05 after Duncan's test. Significance level: *** $p \leq 0.001$; ** $p \leq 0.01$; * $p \leq 0.05$; ns = not significant.

In the first experiment, there was also an increase in carotenoid content in plants treated only with Se in the floating system compared to plants grown without Se and I, and in plants treated with Se + I in aeroponics compared to plant treated only with I or only

with Se. The carotenoid content was also 28.7% higher in aeroponics than in the floating system (Table 3).

In the second experiment, plants grown in aeroponics showed a higher chl a/chl b compared to plants grown in the floating system (Table 4). In plants cultivated without iodine, a significantly lower total phenol content was detected when grown in aeroponics compared to the floating system (Table 4).

**Table 4.** Total phenol, flavonoid content, antioxidant capacity (FRAP), chlorophyll a to b ratio (chl a/chl b), total chlorophyll, and carotenoid contents, measured at harvest, in leaves of lettuce plants grown in two different hydroponic systems (HS), floating (FS) and aeroponics (AE), and at different concentrations of I and Se in the nutrient solution, in the second experiment.

| Hydroponic System | I Added (μM) | Se Added (μM) | Total Phenols (mg GAE g⁻¹ FW) | Flavonoids (mg Catechin g⁻¹ FW) | FRAP (μmol Fe (II) g⁻¹ FW) | chl a/chl b | Chls Tot (mg g⁻¹ FW) | Car (mg g⁻¹ FW) |
|---|---|---|---|---|---|---|---|---|
| FS | 0 | 0 | 3.06 | 1.27 | 27.2 | 2.83 | 1.112 | 0.205 |
|  |  | 13 | 2.95 | 1.41 | 30.0 | 2.66 | 0.908 | 0.160 |
|  | 5 | 0 | 2.97 | 1.39 | 27.3 | 2.58 | 1.065 | 0.176 |
|  |  | 13 | 2.80 | 1.20 | 28.2 | 2.85 | 0.952 | 0.174 |
| A | 0 | 0 | 3.06 | 1.24 | 28.9 | 2.87 | 0.892 | 0.172 |
|  |  | 13 | 2.45 | 0.91 | 23.8 | 2.89 | 0.918 | 0.170 |
|  | 5 | 0 | 2.92 | 1.17 | 25.9 | 2.87 | 1.009 | 0.188 |
|  |  | 13 | 2.93 | 1.26 | 28.1 | 3.22 | 0.838 | 0.170 |
| *Mean effect* | | | | | | | | |
| FS |  |  | 2.94 | 1.32 | 28.2 | 2.73 [b] | 1.009 | 0.179 |
| AE |  |  | 2.84 | 1.14 | 26.7 | 2.96 [a] | 0.914 | 0.175 |
|  | 0 |  | 2.88 | 1.21 | 27.5 | 2.81 | 0.957 | 0.177 |
|  | 5 |  | 2.90 | 1.25 | 27.4 | 2.88 | 0.966 | 0.177 |
|  |  | 0 | 3.00 | 1.27 | 27.3 | 2.79 | 1.019 | 0.185 |
|  |  | 13 | 2.78 | 1.19 | 27.5 | 2.90 | 0.904 | 0.168 |
| FS | 0 |  | 3.01 [a] | 1.34 | 28.6 | 2.75 | 1.010 | 0.182 |
|  | 5 |  | 2.88 [ab] | 1.29 | 27.8 | 2.72 | 1.008 | 0.175 |
| AE | 0 |  | 2.76 [b] | 1.07 | 26.4 | 2.88 | 0.905 | 0.171 |
|  | 5 |  | 2.93 [ab] | 1.22 | 27.0 | 3.04 | 0.923 | 0.179 |
| FS |  | 0 | 3.01 | 1.33 | 27.3 | 2.71 | 1.088 | 0.191 |
|  |  | 13 | 2.87 | 1.30 | 29.1 | 2.76 | 0.930 | 0.167 |
| AE |  | 0 | 2.99 | 1.21 | 27.4 | 2.87 | 0.950 | 0.180 |
|  |  | 13 | 2.69 | 1.08 | 26.0 | 3.05 | 0.878 | 0.170 |
|  | 0 | 0 | 3.06 | 1.25 | 28.1 | 2.85 | 1.002 | 0.189 |
|  |  | 13 | 2.70 | 1.16 | 26.9 | 2.78 | 0.913 | 0.165 |
|  | 5 | 0 | 2.94 | 1.28 | 26.6 | 2.72 | 1.037 | 0.182 |
|  |  | 13 | 2.86 | 1.23 | 28.2 | 3.03 | 0.895 | 0.172 |
| *Significance* | | | | | | | | |
| HS |  |  | ns | ns | ns | *** | ns | ns |
| I |  |  | ns | ns | ns | ns | ns | ns |
| Se |  |  | ns | ns | ns | ns | ns | ns |
| HS × I |  |  | * | ns | ns | ns | ns | ns |
| HS × Se |  |  | ns | ns | ns | ns | ns | ns |
| I × Se |  |  | ns | ns | ns | ns | ns | ns |
| HS × I × Se |  |  | ns | ns | ns | ns | ns | ns |

Means ($n$ = 3) flanked by the same letter are not statistically different for $p$ = 0.05 after Duncan's test. Significance level: *** $p \leq 0.001$; * $p \leq 0.05$; ns = not significant.

### 3.4. Qualitative Characteristics of Leaves during Post-Harvest Storage

The cultivation system did not significantly affect the post-harvest quality of the lettuce leaves, except for weight loss (Table S16), which was 23.1% lower in leaves of lettuce plants grown in aeroponics compared to the floating system after four days of storage (Figure S1, in Supplementary Materials). The effect of the post-harvest storage time, and of the addition of selenium and iodine on the quality parameters, are thus reported separately for the two cultivation systems.

#### 3.4.1. Floating System

Plants grown in floating showed a decreased leaf chlorophyll content during storage at T2 (two days of storage) and at T7 (seven days of storage), compared to T0 (harvest) (Figure 1A). There was also a reduction in leaf carotenoid content at T7 (Figure 1B), and fresh weight from T2 to T7 (Table S12).

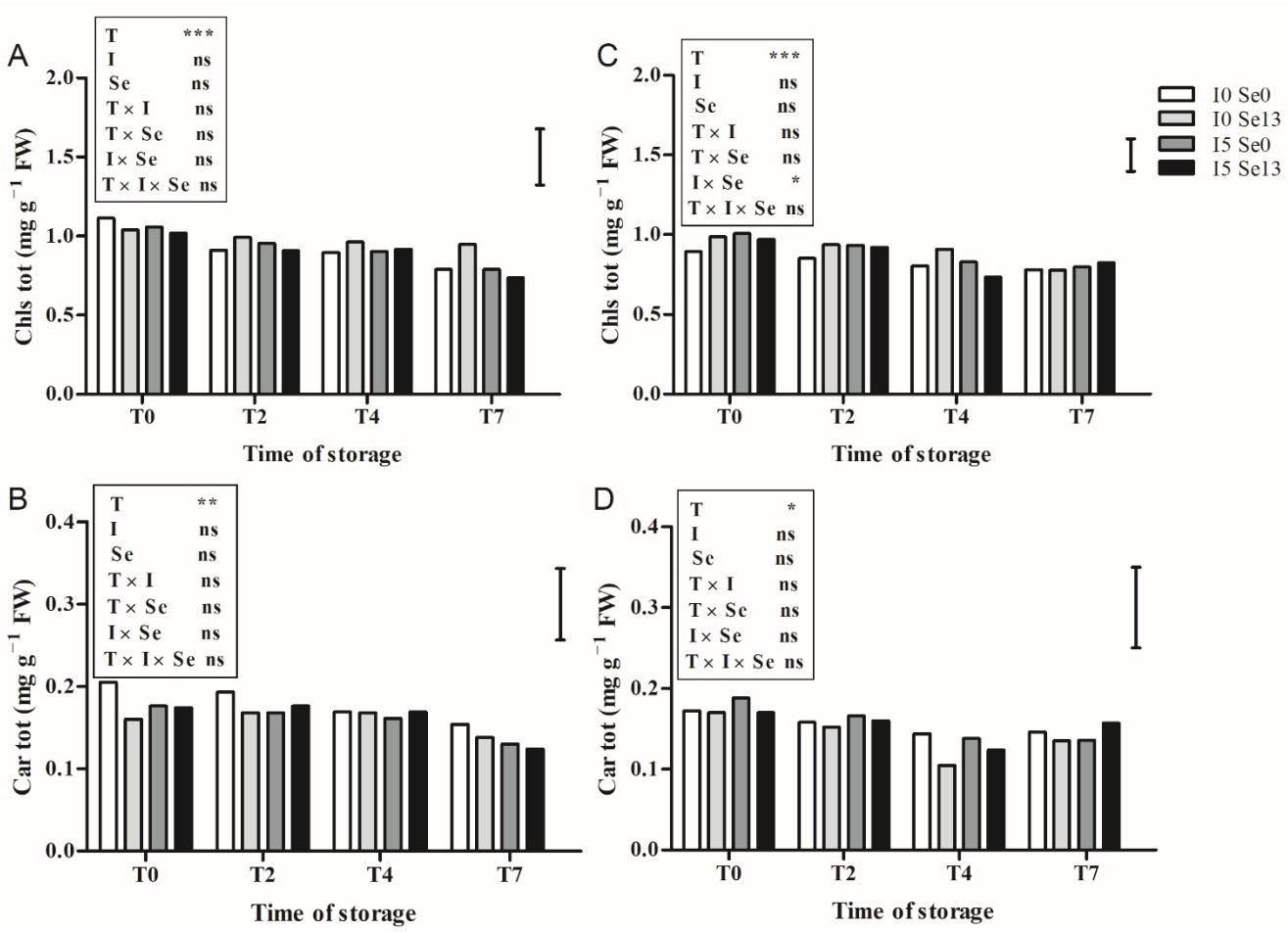

**Figure 1.** Total chlorophyll, FS (**A**) and AE (**C**), and carotenoid, FS (**B**) and AE (**D**), contents measured after 2 (T2), 4 (T4), and 7 (T7) days of storage in leaves of lettuce plants grown at different concentrations of I (0 and 5 μM) and Se (0 and 13 μM) in the nutrient solution, in the second experiment. Values are means of the 3 replicates. Significance level: *** $p \leq 0.001$; ** $p \leq 0.01$; * $p \leq 0.05$; ns = not significant. Bar indicates LSD value.

Treatment with Se increased the chl a/chl b ratio at T2 in plants grown without or with I, and at T4 (four days of storage) in plants treated only with Se (Table S12).

Leaf total phenol content increased at T4 and T7, whereas there was only an increase in the flavonoid content and antioxidant capacity at T7 (Figure 2A–C).

### 3.4.2. Aeroponics

There was also a reduction in leaf chlorophyll content during post-harvest storage in plants grown in aeroponics at T4 and T7 (Figure 1C). Plants showed a reduction in leaf carotenoid content at T7 (Figure 1D), and of fresh weight at T2 and T7 (Table S14).

The highest chl a/chl b ratio was detected at T0 in plants treated with Se and I, and the lowest at T4 in plants treated only with Se compared to control (Table S14).

Increases in the leaf total phenol content at T7 and antioxidant capacity at T2 and T7 were detected during post-harvest storage (Figure 2D–F). The antioxidant capacity was higher in plants treated with Se + I compared to plants treated only with Se. Treatment with Se induced a reduction in flavonoid content in plants not treated with I (Figure 2F).

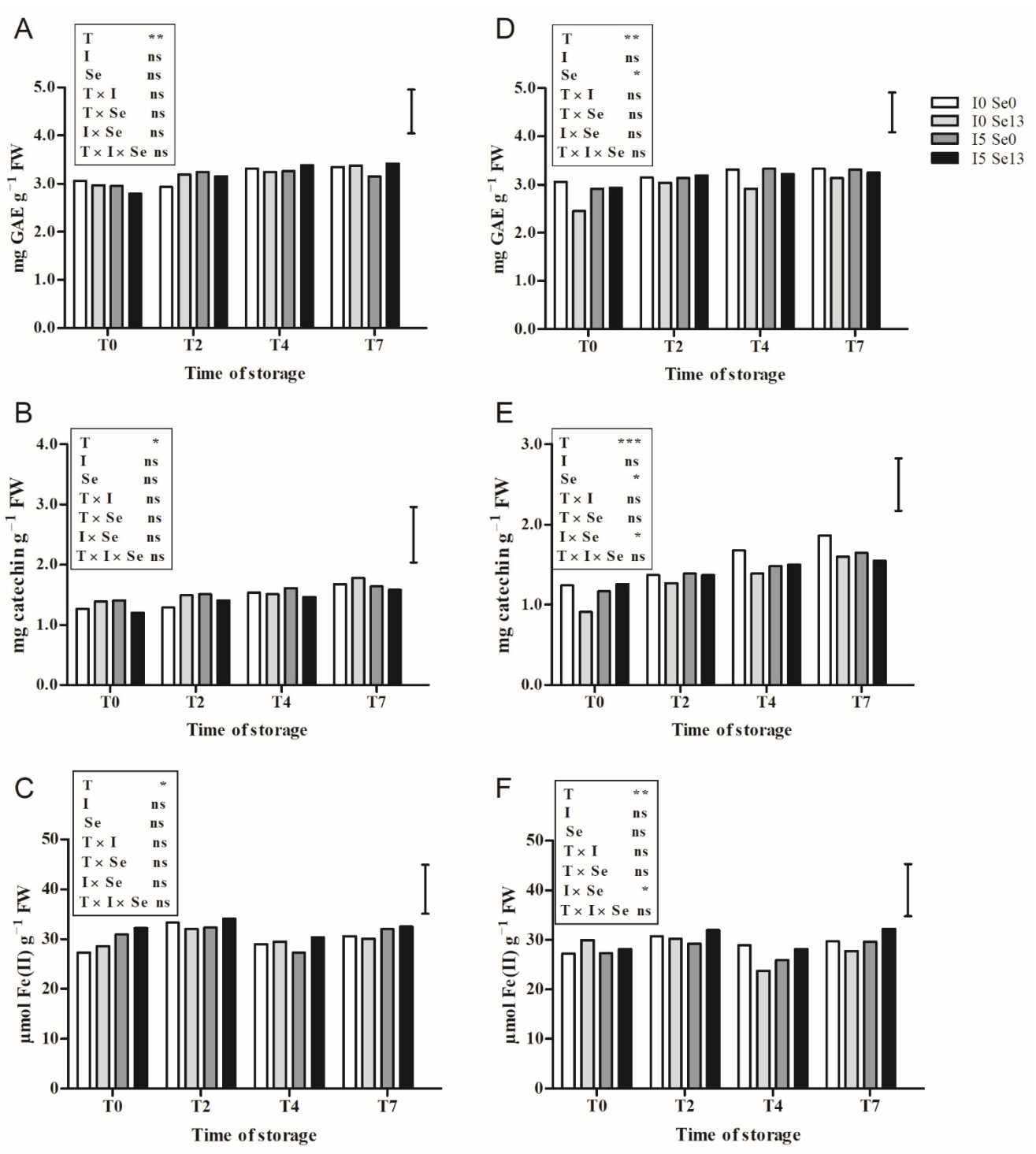

**Figure 2.** Total phenol, FS (**A**) and AE (**D**), and flavonoid, FS (**B**) and AE (**E**), content, and antioxidant capacity, FS (**C**) and AE (**F**), measured after 2 (T2), 4 (T4), and 7 (T7) days of storage in leaves of lettuce plants grown at different concentrations of I (0 and 5 μM) and Se (0 and 13 μM) in the nutrient solution, in the second experiment. Values are means of the 3 replicates. Significance level: *** $p \leq 0.001$; ** $p \leq 0.01$; * $p \leq 0.05$; ns = not significant. Bar indicates LSD value.

## 4. Discussion

### 4.1. Selenium and Iodine Content

Growing plants in a solution enriched with 5 μM of I, supplied as KI, and 13 μM of Se, supplied as $Na_2SeO_4$, resulted in the effective I and Se biofortification of lettuce.

The higher content of iodine detected in I-enriched plants cultivated in the floating system during the second experiment contrasts with previous findings [18], where a higher I uptake was detected in lettuce plants cultivated in aeroponics compared to the floating system. In the present experiment, the higher ratio between leaf area and fresh biomass and the lower ratio between shoots and roots shown by plants grown in floating system (data not shown) could have induced a higher transpiration [35,36] and a higher I uptake per unit of leaf fresh biomass, and thus a higher I accumulation in leaf tissues, as suggested by Smoleń [21], who found that iodine added to the nutrient solution is taken up by roots and transported to the shoot by the xylem.

The higher accumulation of selenium in plants grown in aeroponics compared to those grown in the floating system could be ascribed to the more efficacious nutrient supply to the roots provided by aeroponics as reported by Blok [37].

In I-enriched plants, the leaf iodine content varied between 2.11 and 5.7 mg kg$^{-1}$ FW, and in plants enriched with $Na_2SO_4$, the leaf Se content ranged between 0.10 and 0.21 mg kg$^{-1}$ FW. We calculated that a serving size of lettuce of 100 g, biofortified with I and Se, contained from 211 (floating system, first experiment) to 518 (floating system, second experiment) μg of I and from 10 (floating system, first experiment) to 21 (aeroponics, first experiment) μg of Se. The consumption of 100 g of lettuce could thus satisfy from 141% to 345% and from 14% to 30% of the adequate intake (AI) of I and Se, respectively (Tables S3 and S4).

In all treatments, the health risk index (HRI) of I and Se was always below 1, meaning that the daily human exposure to these amounts of I and Se is not expected to have a hazardous effect over a lifetime [34] (Tables S3 and S4).

Considering the above-mentioned data for biofortified lettuce and that the tolerable upper intake levels of I and Se are 600 and 300 μg day$^{-1}$, respectively [34], it follows that the harmful amount of fresh lettuce leaves consumed per day is 116 g.

### 4.2. Effect of Biofortification on Biomass Production and Leaf Quality

Plant growth and photosynthetic pigment contents were not negatively affected by the addition of 13 μM Se and 5 μM of I to the nutrient solution. This result has been previously found for similar concentrations of Se in spinach [38], lettuce [21,39], and basil [39], as well as of I in broccoli raab, curly kale, mizuna, red mustard [40], basil [18,41], and lettuce [17,18], when the microelements were applied separately. In addition, a higher content of total chlorophylls expressed on a dry weight basis was detected in plants treated with Se during the first experiment. In the literature, an increase of chlorophyll content was previously detected in lettuce and chicory plants treated with 0.5 and 1 mg Se L$^{-1}$ [7].

The increase in fresh biomass observed in the Se-treated plants is consistent with the results of previous studies [42] and clearly indicates that a small amount of selenium (from 6 to 13 μM) might be beneficial for plant growth by increasing the antioxidant metabolism [8,42]. According to our results, phenol content and the antioxidant capacity were not affected by Se and I biofortification. In previous works, total phenol content was not affected by treatments with I in carrot [43] and tomato [44], and by biofortification with Se in basil [45] and carrot [43].

The maximum allowable levels of limits for nitrates in some vegetable species, such as spinach, rocket salad, and lettuce, are set by EU regulation 1258/2011. These limits range between 2000 and 7000 mg kg$^{-1}$ FW, depending on the plant species, growth cycle, and type of cultivation, and are higher for vegetables grown under greenhouses in autumn and winter compared to the open field cultivation during spring–summer [46]. In our work, the nitrate content in lettuce leaves was always considerably lower than the maximum value allowed for lettuce grown in autumn and winter (5000 mg kg$^{-1}$ FW) and was affected by the cultivation system and by the interaction among the cultivation system and the Se concentration in the nutrient solution.

The reduction in nitrate content in leaves of Se-enriched plants is consistent with results obtained in lettuce [8] and corn salad [47]. The reduced leaf nitrate content in

Se-enriched plants grown in aeroponics in our experiment could be due to the higher leaf Se accumulation compared to plants grown in the floating system. This reduction could be ascribed to the capacity of Se to promote nitrate reductase and glutamate synthase activities [8]. However, in other studies, selenium treatments did not affect the nitrate leaf content [48,49]. The lack of an effect of the I treatment on the leaf nitrate content is in agreement with findings in radish [50] and sea fennel [51]. In contrast, other studies have shown that non-toxic concentrations of iodine induced an increase in nitrate content in leaves of spinach [52], broccoli, mizuna [40], and tomato [44]. On the other hand, lower nitrate contents were detected in kale, broccoli, raab, and mizuna treated with 1.5 mg I L$^{-1}$ [40] and in lettuce treated with toxic KI concentrations [17].

The evidence that iodine and selenium did not negatively affect the quality parameters is important for the consumer, meaning that biofortification with these two microelements does not affect the quality of the final product.

### 4.3. Effect of Cultivation System on Biomass Production and Leaf Quality

Lettuce plants showed a significantly lower growth in the floating system than in aeroponics, in accordance with a previous study [18], in which a higher biomass production was detected in lettuce plants grown in aeroponics. In general, the refresh of the nutrient solution at the root surface, and thus a frequent replenishment of the mineral nutrients required for an optimal plant growth, is more efficient in the aeroponics system compared to the floating system, [53]. A dissolved oxygen level of 5 mg L$^{-1}$ is deemed acceptable for many crops cultivated in hydroponics [54]; thus, the lower dissolved oxygen concentration detected in the nutrient solution (about 5 mg L$^{-1}$ against more than 8.5 mg L$^{-1}$ in aeroponics) might be responsible for the growth inhibition detected in floating system. In previous experiments conducted on lettuce grown in hydroponics, a positive correlation was found between the biomass production and the oxygen concentration in the nutrient solution in the range 3–7 mg L$^{-1}$ [55].

Differences in biomass production between the two experiments could have been due to the different environmental conditions, in particular to a lower mean air temperature and cumulative solar radiation in the second experiment (Tables S1 and S2).

The reduction in nitrate content in leaves of lettuce grown in aeroponics is consistent with the results obtained by Puccinelli et al. [18]. The higher nitrate accumulation in leaves of lettuce grown in the floating system may be the result of the reduced plant growth [56]. Despite that, differences between cultivation systems when nitrates are expressed on a dry weight base were detected only in the second experiment (Tables S3 and S4). In the first experiment, the differences were due to the different dry matter content.

The increased chlorophyll and carotenoid contents in plants grown in aeroponics could have been due to a general promotion of the growth and antioxidant system, which may induce an increase in the content of these pigments [47]. A higher content of pigments in the leaves might be valued for a ready-to-eat salad because consumers prefer a darker coloration [57].

### 4.4. Effects of Se and I Biofortification on Lettuce Leaf Quality during Post-Harvest Storage

Lettuce has quite a short post-harvest life, no more than seven days, and the main processes that affect post-harvest quality are water and chlorophyll losses and decay [58].

The reduction in fresh weight recorded in our experiment during storage is consistent with previous results obtained in lettuce plants [59] and is ascribed to the water lost by leaf tissues. In fact, a higher relative water content in lettuce leaves at harvest leads to less water loss during storage [60].

The oxidative loss of chlorophyll and carotenoid content during post-harvest, observed in our experiment, commonly starts a few days after the harvest of lettuce leaves, with a consequent discoloration [58]. It has previously been reported that the chlorophyll content decreased three times slower at 4 °C than at environmental temperatures, when it decreased by almost half in the first few days after harvest [58]. The reduction in chlorophyll content

during storage has been reported in leafy vegetables, such as rocket, chicory, and Swiss chard [61], with a consequent loss of market quality, since greenness is one of the main qualities valued by consumers.

When phenol content was calculated on a dry weight basis, no significant effects of the time of storage or treatments were detected (Tables S10 and S11). The increase in leaf phenol content observed during post-harvest storage can be attributed to a concentration effect because of the water loss.

Se antioxidant properties may be responsible of the increment of the antioxidant capacity in leaves of plant treated with Se [6]. In fact, Se can alleviate oxidative stress, induced by internal or external factors, by improving glutathione peroxidase (GSH-Px) activity [62].

## 5. Conclusions

Our results provided evidence that the simultaneous addition of selenium and iodine at doses of 13 μM and I 5 μM, respectively, increased the content of the two microelements in lettuce leaves without any negative interactions in the plants, and without affecting quality during post-harvest storage. This could lead to the production of leaves biofortified with Se and I, according to specific dietary needs, without altering the quality of the final product.

The consumption of a single 100 g serving of lettuce leaves could satisfy from 140% to 345% and from 14.2% to 30% of the adequate intake (AI) of I and Se, respectively. Moreover, the HRI of I and Se was always below 1.

Both the floating system and aeroponics were found to be suitable cultivation systems for the production of Se and/or I biofortified lettuce. Biofortification with Se was more effective in aeroponics compared to the floating system, whereas I biofortification was more effective in the floating system.

**Supplementary Materials:** The following are available online at https://www.mdpi.com/article/10.3390/horticulturae7120590/s1, Figure S1: Average value of weight loss after 2 (T2), 4 (T4) and 7 (T7) days of storage in leaves of lettuce plants grown in floating system (FS) and aeroponics (AE) at different concentrations of I (0 and 5 μM) and Se (0 and 13 μM) in the nutrient solution, in the second experiment. Bar indicates LSD value, Table S1: Environmental parameters registered during the first experiment, Table S2. Environmental parameters registered during the second experiment, Table S3. Total phenols, antioxidant capacity (FRAP), total chlorophyll, carotenoid contents, measured at harvest, in leaves of lettuce plants grown in two different hydroponic systems (HS), floating (FS) and aeroponics (AE), and at different concentrations of I and Se in the nutrient solution, in the first experiment, Table S4. Total phenol, flavonoid content, antioxidant capacity (FRAP), total chlorophyll and carotenoid contents, measured at harvest, in leaves of lettuce plants grown in two different hydroponic systems (HS), floating (FS) and aeroponics (AE), and at different concentrations of I and Se in the nutrient solution, in the second experiment, Table S5. Se estimated daily intake ($EDI_{Se}$), Se estimated dietary intake expressed as percentage of the Se adequate intake ($EDI_{Se}/AI$), and health risk index (HRI) of a serving (100 g) of lettuce grown in two different hydroponic systems with different concentrations of I and Se in the nutrient solution, Table S6. I estimated daily intake ($EDI_I$), Se estimated dietary intake expressed as percentage of the Se adequate intake ($EDI_I/AI$), and health risk index (HRI) of a serving (26 g) of lettuce grown in two different hydroponic systems with different concentrations of I and Se in the nutrient solution, Table S7. Weight reduction, total phenol, flavonoid content, antioxidant capacity (FRAP), chlorophyll a to b ratio (chl a/chl b), total chlorophyll and carotenoid contents, measured after 2 days of storage, in leaves of lettuce grown in two different hydroponic systems with different concentrations of I and Se in the nutrient solution, Table S8. Weight reduction, total phenol, flavonoid content, antioxidant capacity (FRAP), chlorophyll a to b ratio (chl a/chl b), total chlorophyll and carotenoid contents, measured after 4 days of storage, in leaves of lettuce grown in two different hydroponic systems with different concentrations of I and Se in the nutrient solution, Table S9. Weight reduction, total phenol, flavonoid content, antioxidant capacity (FRAP), chlorophyll a to b ratio (chl a/chl b), total chlorophyll and carotenoid contents, measured after 7 days of storage, in leaves of lettuce grown in two different hydroponic systems with different concentrations of I and Se in the nutrient solution, Table S10. Total chlorophyll, carotenoid,

total phenol, flavonoid content and antioxidant capacity (FRAP) measured at harvest (T0) and after 2 (T2), 4 (T4) and 7 (T7) days of storage, in leaves of lettuce grown in floating system with different concentrations of I and Se in the nutrient solution, Table S11. Total chlorophyll, carotenoid, total phenol, flavonoid content and antioxidant capacity (FRAP) measured at harvest (T0) and after 2 (T2), 4 (T4) and 7 (T7) days of storage, in leaves of lettuce grown in aeroponics with different concentrations of I and Se in the nutrient solution, Table S12. Chlorophyll a to b ratio (chl a/chl b), total chlorophyll, carotenoid contents and weight reduction, measured at harvest (T0), after 2 (T2), 4 (T4) and 7 (T7) days of storage, in leaves of lettuce plants grown in floating system at different concentrations of I and Se in the nutrient solution, Table S13. Total phenol, flavonoid content and antioxidant capacity (FRAP), measured at harvest and after 2, 4 and 7 days of storage, in leaves of lettuce plants grown in floating system at different concentrations of I and Se in the nutrient solution, Table S14. Chlorophyll a to b ratio (chl a/chl b), total chlorophyll, carotenoid contents and weight reduction, measured at harvest and after 2, 4 and 7 days of storage, in leaves of lettuce grown in aeroponics at different concentrations of I and Se in the nutrient solution, Table S15. Total phenol, flavonoid content and antioxidant capacity (FRAP) measured at harvest and after 2, 4 and 7 days of storage, in leaves of lettuce plants grown in aeroponics at different concentrations of I and Se in the nutrient solution, Table S16. Weight loss during post-harvest: results of four-way ANOVA with hydroponic system (HS), selenium (Se), idione (I) and time during post-harvest (T).

**Author Contributions:** Conceptualization, M.P., L.I., F.M. and B.P.; methodology, M.P., L.I., F.M. and B.P.; formal analysis, M.P. and I.R.; investigation, M.P., L.I., F.M. and B.P.; resources, M.P., L.I., F.M. and B.P.; data curation, M.P.; writing—original draft preparation, M.P.; writing—review and editing, M.P. and B.P.; visualization, M.P., L.I., F.M. and B.P.; supervision, M.P., L.I., F.M. and B.P.; project administration, L.I.; funding acquisition, L.I. All authors have read and agreed to the published version of the manuscript.

**Funding:** This work was supported by the project "Aeroponica 2.0, Technological development and testing of an aeroponic system for soilless greenhouse cultivation" POR FSE 2014–2020, Regione Toscana, co-funded by EDO Radici Felici Srl.

**Institutional Review Board Statement:** Not applicable.

**Informed Consent Statement:** Not applicable.

**Data Availability Statement:** The data presented in this study are available within the article and supplementary material.

**Conflicts of Interest:** The authors declare no conflict of interest. The funders had no role in the design of the study; in the collection, analyses, or interpretation of data; in the writing of the manuscript; or in the decision to publish the result.

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
