# Peer review of "Effects of Individual and Simultaneous Selenium and Iodine Biofortification of Baby-Leaf Lettuce Plants Grown in Two Different Hydroponic Systems"

_horticulturae, doi:10.3390/horticulturae7120590_

Round 1
Reviewer 1 Report
I think that the manuscript entitled “Individual and simultaneous selenium and iodine biofortification of lettuce plants grown in two different hydroponic systems" deserves publication in Horticulturae after minor revision. The work is impressive in terms of the number of presented results, although it seem to me that it is better to present in two separate works. The article is interesting for the readers of the journal, the research is scientifically sound and up-to-date. The work is well written and I suggest you only correct a few errors / inaccuracies to improve the quality of your work:
Lines 2-4: suggesting clarify the title so that it corresponds to the presented results.
Line 170: please complete information about the methanol extract
Lines 209: why did you choose Duncan’s test and not Tukeya test which is a more accurate and suitable for such a large amount of data. The results show four values of „P”.
Fig 1. What they refer to is not selected in the interaction windows.
All tables: pleas change “Fresh” into “Fresh weight”; “Dry” into “Dry weight”; dry matter” into “dry weight”
All tables: pleas change “n” into “n”
Table 3, 4, S7-S11, 13, 15, Fig. 3C I 5C: pleas change “µmol” into “µM”
References 89 literature items would indicate that the topi cis known and researched. Moreover, it is review, not a research one.
Whether biofortification affected sensory properties?
Author Response
I think that the manuscript entitled “Individual and simultaneous selenium and iodine biofortification of lettuce plants grown in two different hydroponic systems" deserves publication in Horticulturae after minor revision. The work is impressive in terms of the number of presented results, although it seem to me that it is better to present in two separate works. The article is interesting for the readers of the journal, the research is scientifically sound and up-to-date. The work is well written, and I suggest you only correct a few errors / inaccuracies to improve the quality of your work:
Lines 2-4: suggesting clarify the title so that it corresponds to the presented results.
The title has been modified.
Line 170: please complete information about the methanol extract
Information about methanol extract have been added to the text.
Lines 209: why did you choose Duncan’s test and not Tukey test which is a more accurate and suitable for such a large amount of data. The results show four values of „P”.
Duncan’s test is one of the several methods for multiple comparison, and we have published several papers using this method. However, we thank the Reviewer for her/his suggestions and we will use the Tukey test to analyze the data of future experiments.
Fig 1. What they refer to is not selected in the interaction windows.
Data previously shown in Figure 1 are now reported in Tables 1 and 2.
All tables: pleas change “Fresh” into “Fresh weight”; “Dry” into“Dry weight”; dry matter” into “dry weight”
Text has been modified.
All tables: pleas change “n” into “n”
Text has been modified.
Table 3, 4, S7-S11, 13, 15, Fig. 3C I 5C: please change “μmol” into “μM”
Only the ferric reducing ability of plasma (FRAP) is expressed in µmol Fe (II) g-1 FW.
References 89 literature items would indicate that the topic is known and researched. Moreover, it is review, not a research one.
26 references have been deleted; text has been corrected
Whether biofortification affected sensory properties?
Unfortunately, we did not perform a sensory evaluation of the biofortified leaves in this study. We thank the Reviewer for their suggestion, and we will include this analysis in the next trials.
Reviewer 2 Report
In the present manuscript, the effects of Se and I biortification on plant growth, leaf quality and leaf I and Se accumulation of baby-leaf lettuce plants grown in a floating and aeroponics system and was investigated. Moreover, the effects of Se and I biofortification on post-harvest quality was also investigated.
Comments:
A conclusionary sentence should be added in the end of the Abstract section.
It is not clear how the experiment was replicated, since only four aeroponic and eight floating systems were used. According to the expwerimental layout, there were eight different treatments which means that they were not sufficiently replicated. For example the authors mentioned in Lines 123-124 that they applied a randomised design which is not possible with the suggested number of replicates, especially in the aeroponics system.
The results in Figures should be presented according to the highest order of interaction of the tested factors. Considering that there was no third order interaction between the tested factors, the Figure would easier to read with this way of presentation. Latin letters should be also used to indicate significant differences when present. However, I believe that the use of a Table (Tables S3 and S4) should better present the results.
According to Table S16, the four-way ANOVA showed significant effects only in the case of HS, T and T x I. However it is not clear to which parameter it refers. Moreover, the presented results in Figures 2-5 show significant effects for other main effects or interactions (e.g. Figure 5 shows a significant effect of Se and Se x I).
The results for estimated daily intake, estimated dietary intake and health risk index were not cited in the Results section although they are discussed in the Discussion section. Moreover, the suggested serving size (26 g) refers to a "five a day" serving size. So, the average intake of vegetables in total shoule be 86.2 g per day. I suggest the calculation should be done on a 100 g basis because the serving size and the amount of servings per day differs between countries (e.g. in Mediterranean countries as Itale where authors come from the daily consumption of vegetables is very high).
The differences in Se and I uptake between the the two systems were not adequally discussed. The reason why Iodine was higher in floating system should exist in the case of Se and vice versa. So, another explanation should be given.
Lines 392-403: a clear conclusion should be given regarding the effect of biofortification with Se and I on nitrates accumulation. For example, what was the reason in the studies where Se biofortification did not affect nitrate accumulation (Lines 392-398)? Why Iodine addition did not affect or affected nitrates content (Lines 398-403)?
Lines 408-410: the number of plant grown in each system is irrelevant since the authors mention the same plant density in both systems (Lines 103-104). Lines 410-413: The differences in dissolved oxygen concentration is also irrelevant since the studied systems differ in their whole concept and not only in the aeration of nutrient solution. The cited studies compare different aeration in the same system.
Lines 426-428: in order to be sure for this argument you have to calculate nitrates content on a dry weight basis. The higher nitrate content in AS should be attributed to higher DM content in leaves. Check the same thing for carotenoids.
Lines 443-450: color determination would help to justify these arguments.
Lines 451-457: Did the authors used intact or minimally processed leaves for the storage experiment? The authors should check whether the increase in total phenols content is attributed to a concentration effect because of the water loss. Calculations on a dry weight basis would help for that purpose.
Author Response
In the present manuscript, the effects of Se and I biortification on plant growth, leaf quality and leaf I and Se accumulation of baby leaf lettuce plants grown in a floating and aeroponics system and was investigated. Moreover, the effects of Se and I biofortification on post-harvest quality was also investigated.
Comments:
A conclusionary sentence should be added in the end of the Abstract section.
A conclusionary sentence has been added to the Abstract.
It is not clear how the experiment was replicated, since only four aeroponic and eight floating systems were used. According to the experimental layout, there were eight different treatments which means that they were not sufficiently replicated. For
example the authors mentioned in Lines 123-124 that they applied a randomised design which is not possible with the suggested number of replicates, especially in the aeroponics
system.
The text has been modified to clarify the experimental design.
The results in Figures should be presented according to the highest order of interaction of the tested factors. Considering that there was no third order interaction between the tested factors, the Figure would easier to read with this way of presentation.
Latin letters should be also used to indicate significant differences when present. However, I believe that the use of a Table (Tables S3 and S4) should better present the results.
The results shown in Figures 1 are now presented in Tables 1 and 2.
According to Table S16, the four-way ANOVA showed significant effects only in the case of HS, T and T x I. However it is not clear to which parameter it refers. Moreover, the presented results in Figures 2-5 show significant effects for other main effects or interactions (e.g. Figure 5 shows a significant effect of Se and Se x I).
Table S16 refers to the ‘weight loss’, that is not reported in plot. This information has been added in the caption to Table S16.
The results for estimated daily intake, estimated dietary intake and health risk index were not cited in the Results section although they are discussed in the Discussion section. Moreover, the suggested serving size (26 g) refers to a "five a day" serving size. So, the average intake of vegetables in total shoule be 86.2 g per day. I suggest the calculation should be done on a 100 g basis because the serving size and the amount of servings per
day differs between countries (e.g. in Mediterranean countries as Itale where authors come from the daily consumption of vegetables is very high).
The EDI and HRI have been calculated using 100 g as serving size, and the discussion has been improved according to the new data.
The differences in Se and I uptake between the two systems were not adequally discussed. The reason why Iodine was higher in floating system should exist in the case of Se and vice versa. So, another explanation should be given.
The differences in Se and I uptake have been discussed in paragraph 4.1.
Lines 392-403: a clear conclusion should be given regarding the effect of biofortification with Se and I on nitrates accumulation.
For example, what was the reason in the studies where Se biofortification did not affect nitrate accumulation (Lines 392-398)?
In our work, the nitrate content in lettuce leaves was affected by the cultivation system and by the interaction among the cultivation system and the Se concentration in the nutrient solution. Discussion of these data are reported at lines 382-388 and 413-417 (see the revised version of the manuscript).
Why Iodine addition did not affect or affected nitrates content (Lines 398-403)?
It is not clear why iodine addition did not affect the nitrates content. Previous studies reported contrasting results as reported at lines 388-393 (see the revised version of the manuscript); this issue would be worth exploring further.
Lines 408-410: the number of plant grown in each system is irrelevant since the authors mention the same plant density in both systems (Lines 103-104).
Text has been corrected.
Lines 410-413: The differences in dissolved oxygen concentration is also irrelevant since the studied systems differ in their whole concept and not only in the aeration of nutrient solution. The cited studies compare different aeration in the same system.
The cited studies that compared different aeration in the same system have been removed.
In order to discuss the possible differences among the two hydroponics systems, in addition to the dissolved oxygen level, some information about the replenishment of nutrients have been added.
Lines 426-428: in order to be sure for this argument you have to calculate nitrates content on a dry weight basis. The higher nitrate content in AS should be attributed to higher DM content in leaves. Check the same thing for carotenoids.
We have calculated the nitrate, chlorophyll, and carotenoid content on a dry weight basis (Table S3, S4). Text in the Discussion Section has been corrected.
Lines 443-450: color determination would help to justify these arguments.
We did not perform the color determination of leaves. We agree with the Reviewer that this parameter could have helped in explaining part of the results. This could be an interesting suggestion for a future work.
Lines 451-457: Did the authors used intact or minimally processed leaves for the storage experiment? The authors should check whether the increase in total phenols content is
attributed to a concentration effect because of the water loss. Calculations on a dry weight basis would help for that purpose.
For the storage experiment we used intact leaves.
The results calculated on a dry weight basis are reported in Table S10 and S11. The text has been corrected according to these results.
Reviewer 3 Report
This is an extremely interesting manuscript about Se and I biofortification of lettuce plants. There are some points in the material and methods section that need to be clarified:
Paragraph 2.1: You say that "During the experiments, the nutrient solution was never replaced or reintegrated." (line 115). How do you know that nutrients' amounts are sufficient to support plants' growth? Moreover, you say that "The pH and EC were 108 checked every day and remained within 10% of the values measured in the newly pre-109 pared nutrient solution." (lines 108-110). How did you practically do that?
Paragraphs 2.3, 2.4.2, 2.4.3: Please specify which leaves of the plants were taken as samples for each determination and why did you choose those specific leaves.
Author Response
This is an extremely interesting manuscript about Se and Ibiofortification of lettuce plants. There are some points in thematerial and methods section that need to be clarified:
Paragraph 2.1: You say that "During the experiments, the nutrient solution was never replaced or reintegrated." (line 115). How do you know that nutrients' amounts are sufficient to support plants' growth? Moreover, you say that "The pH and EC were 108 checked every day and remained within 10% of the values measured in the newly pre-109 pared nutrient solution."(lines 108-110). How did you practically do that?
We have used the same nutrient solution that we have used for the cultivation of leafy vegetables in other experiments, similar to the present trial in term of duration and plant density. In addition, we have analyzed the composition of the nutrient solution at the end of the experiment and the final nutrient composition was still adequate for the cultivation of lettuce plants.
We measured the pH and EC, directly in the systems, by portables devices.
Paragraphs 2.3, 2.4.2, 2.4.3: Please specify which leaves of the plants were taken as samples for each determination and why did you choose those specific leaves.
We used the last two fully expanded leaves. We chose those because they had an average age compared to the leaves of the whole plant.
Reviewer 4 Report
Dear authors,
Major comments:
The present paper investigates the effects of individual and simultaneous selenium and iodine biofortification of lettuce plants grown in two different hydroponic systems. The topic is interesting and significant since, as stated by the authors, there is a lack of information about effects of this simultaneous selenium and iodine biofortification on baby-leaf lettuce. The paper is generally well written, but presentation of the results is very poor. Also, there are several suggestions and issues that authors should consider and improve. My opinion is that the present paper could be published in Horticulturae after major revision and consideration of listed comments.
Specific comments and suggestions:
In my opinion it would be good to put the word baby-leaf in the title.
Abstract
Overall the Abstract is to extensive and to generally written. Please reduce it and give in it the most important results and conclusions.
Introduction
L48 I think that you missed here the word baby-leaf or?
Please give some informations about the effects of biofortification of lettuce leaf quality from previous research papers.
Materials and methods
Did you test the data for homogeneity of variances?
Please add the Se and I transference factor!
Results
The graphs are too small. Please indicate in graph what is HS and also please put letters above graphs for Duncan’s test. I do not understand why are there two separate marks for significance on each graph?
All Tables are to complicated and it is really difficult to follow the results! Please make new tables! Why are there no letters for Duncan’s test on each value in the table?
In figures 2-5 there are no visible differences between different groups, maybe try instead of lines to use columns and merge these figures in one plot of figures. Also put the a,b,c for Duncan’s test in graphs.
Discussion
In L372 and 381 you cannot write about toxicity to plants when you did not measure neither one parameter of oxidative stress like lipid peroxidation, ROS or hydrogen peroxide!
Please add some references to discuss the results in 4.1.!
L435 is not true, the treatment has significant impact on some leaf quality parameters. You did not discuss about significant I*Se impact on Chls content. Also, you did not write about Se*I impact on flavonoid and total antioxidative activity! How do you explain decline in Chl content in T4 and then an increase in T7?
Author Response
The present paper investigates the effects of individual and simultaneous selenium and iodine biofortification of lettuce eplants grown in two different hydroponic systems. The topic is interesting and significant since, as stated by the authors, thereis a lack of information about effects of this simultaneous selenium and iodine biofortification on baby-leaf lettuce. The paper is generally well written, but presentation of the results is very poor. Also, there are several suggestions and issues that authors should consider and improve. My opinion is that the present paper could be published in Horticulturae after major revision and consideration of listed comments.
Specific comments and suggestions:
In my opinion it would be good to put the word baby-leaf in the title.
DONE
Abstract
Overall the Abstract is to extensive and to generally written. Please reduce it and give in it the most important results and conclusions.
Abstract has been improved.
Introduction
L48 I think that you missed here the word baby-leaf or?
We are sorry, but we looked at L48 and we did not find how to incorporate the word baby-leaf in the sentence.
Please give some informations about the effects of biofortification of lettuce leaf quality from previous research papers.
Some information about the effects of the biofortification of lettuce on leaf quality have been added in the Introduction.
Materials and methods
Did you test the data for homogeneity of variances?
Yes, this information has been added to the text.
Please add the Se and I transference factor!
This study aimed at investigating the effects of Se and I biofortification on leaf quality and leaf I and Se accumulation in plants cultivated in two different hydroponic systems, and we did not measure the selenium and iodine concentration in roots. The translocation factor has been determined in a previous work (Puccinelli et al Scientia Hort 2017) aimed at studying the uptake and partitioning of selenium in basil plants.
Results
The graphs are too small. Please indicate in graph what is HSand also please put letters above graphs for Duncan’s test. I do not understand why are there two separate marks for significance on each graph?
Data previously shown in Figure 1 are now reported in Tables 1 and 2. The other figures have been improved.
All Tables are to complicated and it is really difficult to follow theresults! Please make new tables! Why are there no letters forDuncan’s test on each value in the table?
Tables have been modified, and we hope that now they are easier to read.
The letters have been added only when the interaction was statistically significant.
In figures 2-5 there are no visible differences between different groups, maybe try instead of lines to use columns and merge these figures in one plot of figures. Also put the a,b,c for Duncan’s test in graphs.
The plots have been changed in histograms and we have merged the plots in one plot for pigments and one plot for phenols and antioxidant capacity.
The letters have not been added because the highest level of interaction, related to the means reported in the plots, was never statistically significant.
Discussion
In L372 and 381 you cannot write about toxicity to plants when you did not measure neither one parameter of oxidative stress like lipid peroxidation, ROS or hydrogen peroxide!
This part has been removed from the discussion.
Please add some references to discuss the results in 4.1.!
Two references have been added.
L435 is not true, the treatment has significant impact on some leaf quality parameters. You did not discuss about significant *Se impact on Chls content. Also, you did not write about Se*Iimpact on flavonoid and total antioxidative activity!
Text has been corrected.
The effect of Se on the phenols accumulation and antioxidant capacity during storage is discussed at lines 439-445 (see the revised version of the manuscript).
How do you explain decline in Chl content in T4 and then an increase in T7?
Only the mean effect of “time of storage” was statistically significant. There were no statistical differences in chlorophyll and carotenoid content between T4 and T7 (Table S10, S11, S12,S14).
Round 2
Reviewer 2 Report
The authors have addressed my comments. Therefore, I recommend the acceptance of the manuscript.
Author Response
Thank you.
Reviewer 4 Report
Dear authors,
thank you for all your comments and improvements. There are still some minor changes that you should consider and improve before publishing.
L165 there is a sentence in Italian.
L219-L221 please add is it in the first/second experiment or both!
L358 you can not write about the damage of the photosynthetic apparatus when you only measure Chl and Car concentration.
In Figure 1 is LDS value instead of LSD.
Author Response
L165 there is a sentence in Italian.
The sentence has been removed.
L219-L221 please add is it in the first/second experiment or both!
The required information has been added.
L358 you can not write about the damage of the photosynthetic apparatus when you only measure Chl and Car concentration.
The sentence about the damage of the photosynthetic apparatus has been removed from the text.
In Figure 1 is LDS value instead of LSD.
Text has been corrected.